# The fishery performance indicators for global tuna fisheries

Jessica K. McCluney[1], Christopher M. Anderson [2] & James L. Anderson[3]

We characterize the ecological, economic, and community performance of 21 major tuna fisheries, accounting for at least 77% of global tuna production, using the Fishery Performance Indicators. Our analysis reveals that the biggest variations in performance among tuna fisheries are driven by the final markets that they target: international sashimi market tuna fisheries considerably outperform a comparison set of 62 non-tuna fisheries in the Fishery Performance Indicator database, international canned tuna market fisheries perform similarly to the comparison set, and tuna fisheries supplying local markets in coastal states considerably underperform the comparison set. Differences among regional fishery management organizations primarily reflect regional species composition and market access, despite stark variation in governance, management, and other enabling conditions. With a legacy of open access, tuna's harvest sector performance is similar across all fisheries, reflecting only a normal return on the capital and skill invested: industrial vessels slightly outperform semi-industrial and artisanal vessels. Differences emerge in the post-harvest sector however, as value chains able to preserve quality and transport fish to high value markets outperform others.

[1] McCluney Seafood Strategies, PO Box 95196, Seattle, WA 98145, USA. [2] School of Aquatic and Fishery Sciences, University of Washington, 1122 Boat Street, Box 355020, Seattle, WA 98195, USA. [3] Institute for Sustainable Food Systems and Food & Resource Economics, University of Florida, 1741 Museum Road, PO Box 110570, Gainesville, FL 32611, USA. Correspondence and requests for materials should be addressed to C.M.A. (email: cmand@uw.edu)

Well-managed fisheries have the potential to provide food for a growing global population, sustain the livelihoods of hundreds of millions of people, and support the communities who depend on them. However, fisheries often fall short of their full potential to provide sustainable, secure livelihoods, with an estimated US$80 billion per year in unrealized profits[1]. These estimates include reduced catch due to overfishing, excessive harvest cost, low processing yields, product waste, and a failure to reach the highest value markets. The human consequences are lost revenue to small-scale and industrial harvesters and processors[2], foregone high quality protein to consumers[3], and reduced food and income security for fishing dependent communities in both developed and developing regions.

While advances in management worldwide have improved the sustainability of fish stocks in the last decade[4], much of this gain has been limited to fisheries within the national waters of countries with high governance capacity. Tuna fisheries, whose 4.6 million MT volume and US$12.2 in billion landed value[5] are a crucial source of food and trade income for both developed and developing countries, face particular challenges in improving management. Tuna stocks span multiple national jurisdictions, typically including countries with low management and enforcement capacity, and the high seas. They are thus regulated by regional fishery management organizations (RFMOs) that establish harvest control measures within their regional jurisdictions.

The five tuna RFMOs are constituted of the regions' coastal states and representatives of the distant water fishing nations who are most active, and often capture much of the benefit, in the region[6]. The disparate ways in which different parties benefit from the fisheries often lead to rent-seeking, conflicting interests, and allocative disagreements, hindering the consensus required to establish new measures[7,8], in particular those measures which limit access to fishery resources[9,10]. Even when measures are approved, compliance is imperfect because fishing nations are individually responsible for implementation within each distinct regional jurisdiction, but enforcement often relies on voluntary compliance[11] or coastal states to prevail upon their patron fishing nations to impose sanctions against their own fleets, who heavily rely on the very same resource rent to support their national economies[12].

Given these complex challenges, the RFMO system has made considerable progress since its inception following the 1995 Fish Stocks Agreement to the UN Law of the Sea[13]. Biological data on tuna stocks are being more consistently captured, aggregated, and analyzed at the regional level, then applied to inform management[14,15]. In some cases, they have institutionalized incumbent access rights through capacity controls, limited entry, and rights-based systems[16,17]. Further, enforcement is improving though the 2009 Port State Measures Agreement, which authorizes ports to refuse services to vessels not complying with RFMO measures. However, challenges persist in leveraging better biological information and access limitations into changes that would lead to higher profitability, and improvements in the level and distribution of benefits to stakeholder communities.

To improve the generation and distribution of benefits from tuna fisheries, NGOs and global aid agencies have begun partnering with scientists, managers, and industry[18–20]. Stock assessments show significant biological progress[21,22], but high stock levels will not necessarily lead to an economically healthy industry that can support its participating communities if there is no mechanism to capture the full value potential of the harvest[23,24]. Sharing knowledge, information, and experiences across fisheries in order to identify opportunities for triple-bottom line improvements requires a systematic description of biological, economic, and social outcomes, as well as current

management and market structures in the fishery[25]. However, tuna fisheries are like other fisheries in that much of the data necessary to assess social and economic performance are lacking; information on prices, employment, effort, labor conditions, and many other factors is not consistently collected for systematic cross-comparison. It is therefore difficult to characterize who is benefitting from each tuna fishery, and where there might be opportunities for learning or improvement.

This paper captures a baseline snapshot of the performance of the world's major tuna fisheries circa 2012, before the NGO partnerships above take effect. We first identify 21 major fisheries, comprising the major gear groups, markets, and management strategies across all five RFMOs, representing over 77% of global tuna landings[26]. We evaluate the ecological, economic and social performance of each fishery using the Fishery Performance Indicators[27] (FPIs), a rapid assessment instrument comprised of 68 outcome measures, each Likert-scored from a low of 1 to a high of 5, along with 54 measures of governance and management enabling conditions that may affect performance. We compare tuna fisheries with each other and with non-tuna benchmarks on indicators of stock health, harvest sector, and post-harvest sector performance, grouped by final product market, RFMO, and industrial scale of production. We find that the biggest differences in tuna fishery performance arise in the post-harvest sector, and are most associated with value chains that preserve quality and transport to high-value markets; differences among RFMOs are driven primarily by species composition.

## Results

**Characterizing Global Tuna Fisheries.** Table 1 shows the characteristics of the 21 major RFMO-level fisheries, arranged by RFMO and ranked by 2009 catch volume[21,28,29]. (Detailed profiles in Supplementary Table 1.) Since the FPIs are designed to assess fishery management systems, our fishery units-of-analysis consider one or more fleets harvesting the same stocks within the same management jurisdiction under the same set of rules for access and harvest, using similar levels of technology, and supplying similar markets. Global tuna production relies on two primary harvest strategies. Fleets using purse seine, gillnet, and pole & line or trolling gear predominantly target skipjack (Katsuwonus pelamis), as well as yellowfin (Thunnus albacares) and some albacore (Thunnus alalunga), for canned or pouched products in the shelf-stable markets. Fleets using longline and handline gear target, among other large pelagic species, yellowfin, bigeye (Thunnus obesus), albacore, and bluefin (Thunnus orientalis; Thunnus maccoyii) for the fresh and frozen steak and sashimi markets. Both harvest strategies supply a range of products to various local markets, either intentionally or by landing fish that is denied export for poor quality or other reasons. Additionally, a miniscule relative volume of bluefin is trapped or ranched, fattening purse-seine caught juveniles in pens. Within these strategies, fisheries are delineated by RFMO, scale of production[30], and regulations as implemented by the flag states.

The Western-Central Pacific Fishery Commission (WCPFC) establishes harvest recommendations within the western Pacific Ocean, from Japan through southeast Asia and Pacific Island nations down to Australia. Its fisheries include distant water industrial purse seiners from Asia, a similar US fleet, and domestically capitalized fleets in primarily Indonesia and the Philippines. Within the WCPFC region, the Parties to the Nauru Agreement (PNA), a subgroup of eight small island states, complement RFMO regulations for purse seiners with the only functional harvest rights for purse seined tuna. Purse seiners fishing in PNA waters, the most productive skipjack waters in the

**Table 1 Characteristics of 21 global tuna fisheries and their performance outcomes by triple bottom line and by sector**

| Fishery characteristics | | | | | | | Triple bottom line performance | | | Sector performance | |
| --- | --- | --- | --- | --- | --- | --- | --- | --- | --- | --- | --- |
| RFMO | Scale | Gear | Tuna species | Flag states of fleets | Est. volume (000 Tons 2009) | Regional management controls | Ecology | Economics | Community | Harvest | Post-harvest |
| WCPFC | I | DWPS | SJ/YF | JPN, TWN, KOR | 935 | VD caps; seasons | 3.60 | 3.68 | 4.00 | 3.83 | 3.88 |
| | I | PS | SJ/YF | IDN, PHL | 551 | Seasons | 3.13 | 3.56 | 3.98 | 3.65 | 3.77 |
| | I | PS | SJ/YF | PNG, SLB, MHL, FSM | 361 | FSMA VD caps; seasons | 4.20 | 3.48 | 3.15 | 3.34 | 3.24 |
| | I | DWPS | SJ/YF | USA | 282 | Treaty VD caps; seasons | 3.38 | 3.43 | 3.53 | 3.54 | 3.62 |
| | I | DWLL | YF/BF | JPN, TWN, KOR | 151 | Vessel limits; GHLs | 3.25 | 3.88 | 4.04 | 3.62 | 4.37 |
| | I | LL | YF/BE/BF | JPN, TWN | 65 | Vessel limits; GHLs | 3.13 | 3.98 | 4.22 | 3.62 | 4.64 |
| | A | LL/HL | YF/BE | IDN, PHL | 38 | None | 3.50 | 3.58 | 3.83 | 3.41 | 3.74 |
| | I | LL | AL/YF | FJI | 14 | Closed areas | 4.25 | 3.74 | 3.21 | 3.20 | 3.80 |
| IOTC | SE | GN/LL | SJ | LKA, YEM, OMN, PAK, IRN, IND, COM | 303 | None | 3.75 | 2.96 | 3.64 | 3.35 | 2.93 |
| | I | DWPS | SJ | EU (FRA, ESP) | 243 | Closed areas | 3.75 | 3.85 | 4.12 | 4.13 | 3.87 |
| | SE | LL/HL | YF | LKA, MDV, IDN, SYC, YEM, OMN, PAK, IRN, IND, COM | 119 | None | 4.16 | 3.67 | 3.99 | 3.54 | 3.96 |
| | SE | PL | SJ | MDV | 110 | None | 4.75 | 3.54 | 3.93 | 4.00 | 3.38 |
| IATTC | I | PS | SJ | ECU, COL, VEN, PAN, NIC | 454 | Capacity limits; seasons | 3.63 | 3.83 | 4.29 | 3.89 | 4.10 |
| | I | PL/TR | AL | USA, CAN | 41 | GHLs | 5.00 | 4.08 | 4.36 | 4.12 | 4.34 |
| | A | LL | YF/BE | ECU, PER | 11 | BE TACs | 3.50 | 3.40 | 3.90 | 3.03 | 3.99 |
| | I | PS/RN | BF | MEX | 10 | None | 2.75 | 4.17 | 4.62 | 4.10 | 4.65 |
| ICCAT | I | PS | SJ/YF | EU (FRA, ESP), GHA | 198 | GHLs | 3.63 | 3.76 | 3.76 | 3.99 | 3.71 |
| | I | DWLL | BE/YF/BF | JPN, TWN, KOR | 82 | Vessel cap; TAC | 2.50 | 3.95 | 4.03 | 3.61 | 4.38 |
| | I | PS/TP | BF | ESP, TUR, MLT, HRV, GRC, ITA | 7 | TAC; season; capacity limits | 2.25 | 4.06 | 4.26 | 3.84 | 4.52 |
| CCSBT | I | DWLL | SB | JPN, TWN | 5 | TAC | 3.13 | 3.86 | 4.05 | 3.24 | 4.70 |
| | I | PS/RN | SB | AUS | 4 | TAC | 2.88 | 4.14 | 4.14 | 3.83 | 4.47 |

Scale values are I-industrial, SE-semi-industrial and A-artisinal; gear types are PS- purse seine, LL-longline, HL-handline, GN-gillnet, PL-pole & line, TR-trolling, TP-trapping and RN-ranching (DW-distant water); target species are SJ-skipjack, YF-yellowfin, BE-bigeye, AB-albacore, and BF-bluefin; management approaches are VD-vessel day caps, GHL-guideline harvest levels, TAC-hard total allowable catch. Italicized flag state fleets were directly scored; non-italicized fleets are similar enough to the scored fleets that scores apply to them as well. Directly scored fleets account for at least 60% of each fishery's total volume except for the diffuse, information-poor IOTC SJ GN/LL fishery, where the scored Sri Lankan fleet accounts for an estimated 10% of catch. See Supplementary Table 1 for additional details

world, must hold fishing days granted by PNA countries through one of three avenues[17]. The Asian distant water nations negotiate their number of days bilaterally with PNA countries over price, terms for hiring regional crew, and regional landings shares, while the US distant water fleet bargains multilaterally under what is known as the Vessel Day Scheme (VDS). Fish from both Asian and US purse seine fleets are primarily canned in Thailand. A third avenue for purse seine harvest rights is for the domestic fleets of the Federated States of Micronesia Arrangement (FSMA); these parties receive a priority allocation of fishing days, and land domestically or regionally. The sale of purse seine fishing days has increased government revenue, employment, and processing activity throughout the PNA countries, and refining and expanding this system is a focus of tuna-based development investments. The WCPFC fisheries also include longline fleets targeting mature yellowfin, bigeye, and bluefin for the fresh and frozen Japanese and western markets under RFMO-level vessel limits intended to keep within guideline harvest levels. Lastly, coastal states have artisanal longline and/or handline fleets, whose vessels are too small for international reporting requirements, that land domestically with limited access to the international fresh and frozen market.

The Indian Ocean Tuna Commission (IOTC) manages stocks from the eastern coast of Africa to the western coast of Indonesia. Its fisheries include European distant water purse seiners that land primarily to canneries in Seychelles; domestic-flagged semi-industrial fleets from several IOTC states with hybrid gillnet and longline gear, fishing primarily for local markets; artisanal handline and longline fleets from several IOTC states with access to the international fresh and frozen markets; and the MSC-certified Maldives pole-and-line skipjack fishery. The IOTC has the highest proportion of small-scale catch of all the RFMOs, estimated to account for 40–50% of the region's landings. Reporting from these vessels is poor, leading to greater uncertainty in assessments, catch, and capacity estimates (Gillett 2011).

The Inter-American Tropical Tuna Commission (IATTC) manages stocks in the eastern Pacific Ocean, extending to the

coasts of North, Central, and South America. During our study period, over 90% of the region's landings were caught by domestic-flagged industrial purse seiners in the region who have been allocated capacity, and landed to canneries in Ecuador and Mexico. Other fisheries include the US MSC-certified troll-caught albacore fleets harvesting for US and European canneries; Mexican bluefin ranching for the sashimi market; and domestic artisanal longline fleets known regionally as *fibras*, who target tunas, mahi-mahi, and sharks delivering at sea to semi-industrial longline vessels carrying ice and supplies, and ultimately going to local or international frozen markets. The IATTC was the first formally organized tuna RFMO following the Tuna Stocks Agreement, and is recognized as having the strongest RFMO-level management measures.

The International Commission for the Conservation of Atlantic Tunas (ICCAT) manages stocks in the Atlantic Ocean. Its fisheries include European distant water purse seine fleets fishing off the coast of west Africa in conjunction with Ghanaian bait boats and delivering into Abidjan; Asian distant water longline fleets transshipping to container vessels destined for Asian ports; and the Mediterranean bluefin fleets purse seining or trapping for European markets. Due especially to coverage of the collapse of high-value Atlantic and Mediterranean bluefin, ICCAT has received considerable external scrutiny of its management measures.

Lastly, the Commission for the Conservation of Southern Bluefin Tuna (CCSBT) manages southern bluefin tuna across the Southern Ocean and into the southern latitudes of the Pacific, Atlantic and Indian Oceans, overlapping geographically with the other four RFMO management areas. Its fisheries include the primarily Japanese distant water longliners transshipping to Japan, and Australian bluefin ranching operations for high-value export. CCSBT has the fewest voting members, gear types, and active fishing nations, resulting in a comparatively simplified management framework of stringent total allowable catch limits for its single species.

**Triple bottom line performance**. The right columns of Table 1 summarize the performance outcomes for each fishery. The FPI measures can be combined for interpretation from two perspectives: first, metrics are partitioned by their triple bottom line performance indicators of Ecology, Economics, and Community, then the metrics are re-partitioned to be viewed through Harvest and Post-harvest Sector performance indicators (Supplementary Figure 1).

The Ecology indicator captures the health of tuna stocks, which primarily reflect life histories. Aside from fisheries targeting overexploited bluefin tuna species, Ecology scores range from a low of 3.13 to a high of 5.00. Stock assessments conclude biomass is above or close to BMSY. Differences in Ecology performance are driven by selectivity, where bycatch of juveniles of non-target tuna species, unmarketable billfish, sharks or marine mammals reduces performance. Bluefin scores range 2.25–2.88. Mature yellowfin and bigeye experience significant fishing pressure due to their high value, but pursuing them is more expensive, which prevents severe overfishing. Skipjack is highly productive and globally abundant, resulting in very high Ecology scores across all fisheries, but purse seining with fish aggregating devices leads to tuna bycatch of juvenile yellowfin and bigeye, which are less abundant and decrease those skipjack fisheries' scores.

Economic performance is associated with target species: skipjack is a high-volume, low value commodity product, while others are potentially higher valued, with differences among them arising from individual fisheries' ability to transport products to high paying markets. The highest performers (e.g. Mexican and

Australian bluefin, US albacore, followed by Asian distant water fleets) are longline and handline fleets with the infrastructure to preserve quality. Artisanal fleets, if supplying only local markets, tend to perform like low-value skipjack fleets. In many cases, fisheries with low ecological scores are among the highest economic performers.

It is difficult to identify patterns in community performance. Scores reflect the opportunities and services available to families of fishery participants in their home countries. As a result, these scores often average over outcomes in disparate locations and have lower confidence, especially for distant water fleets. IATTC stands out as a region because their fish is processed within the region. Therefore, fishing benefits are more concentrated in local communities. Most other skipjack is processed in global canning hubs such as Bangkok or Port Victoria, so processing benefits do not accrue to the people the FPIs consider as part of the fishery community. The lower scores of domestic fleets (e.g. in the WCPFC) reflect lower rent generation that supports the fishing community.

Because individual tuna fisheries provide benefits to many geographies and have different structures for generating benefits in harvesting and processing, the triple bottom line partitioning obscures the most interesting differences. Therefore, to be better understand whether tuna fisheries are generating benefits, and to whom they accrue, this analysis explores tuna fishery performance through the sector partitioning of the FPIs.

**Harvest and post-harvest sector outcomes**. The final product market each fishery feeds does not generate performance differences on harvest sector dimension scores (orange outer ring in Fig. 1a) but does dramatically affect performance in the post-harvest sector (beige outer ring). This indicates the post-harvest sector may present better opportunities for capturing additional fishery benefits than the harvest sector. Overall, the tuna harvest sector performance is comparable to the average of the 62 non-tuna fleets in the FPI database (Supplementary Table 2), largely reflecting that tuna fishing jobs are not elevating fishermen beyond comparably skilled jobs in other fisheries. The canned fleet has slightly higher scores, associated with industrial capital investment, higher skills required to captain vessels, and often-lower opportunity cost of internationally sourced crew. This outcome also reflects that ex-vessel tuna markets are commoditized, so individual harvest fleets are not able to influence their own pricing[31,32]. This is even observed among MSC-certified skipjack fleets (e.g. Maldives pole-and-line), who expressed that export canneries do not pay a premium for their MSC-certified raw material.

In the post-harvest sector dimensions, canned products outperform the non-tuna FPI average, and fresh/frozen tuna for high value sashimi products performs even better. Sashimi market processing firms are smaller in scale than canneries, but have more specialized wholesale buyers who are responsive to minor variations in quality and handling, hence the managerial skill required is comparable to capital-intensive canneries, and the cutting skill required exceeds that for canneries. Local market fleets fall below the FPI average, selling into low value markets through product forms that generate lower returns for their processing owners and workers. Since they are often landing sashimi market species, these outcomes suggest that handling, processing, and marketing are an important source of additional fishery benefits, especially where potentially sashimi-grade products are falling into a local market due to poor cold chains, high transportation costs, or environmental compliance issues.

The enabling conditions (Fig. 1b; Supplementary Figure 2) of the international market (both canned and sashimi fleets) are

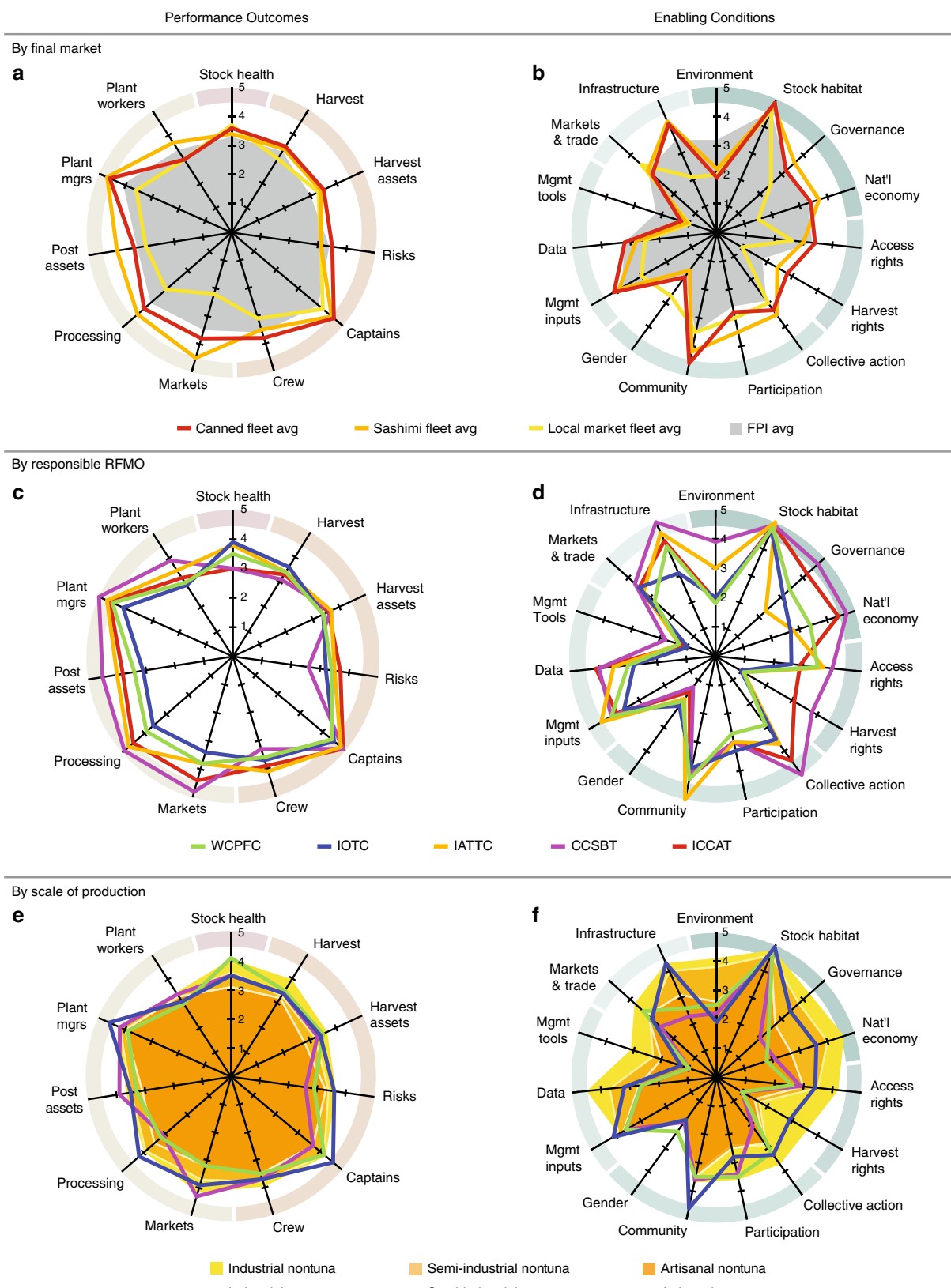

**Fig. 1** Volume-weighted averages of tuna fishery FPI scores. Fisheries are grouped by by final product market (Output panel **a**; Input panel **b**) with a comparison to all non-tuna FPI case studies (grey solid field), governing RFMO (outcomes panel **c**; enabling conditions panel **d**), and industrial scale of production (outcomes panel **e**; enabling conditions panel **f**) with a comparison to non-tuna FPI case studies (yellow solid fields). Radial axes reflect volume-weighted average of the dimension score, with higher scores interpreted as better performance only in the outcome panel. On the outcome panels, a pink outer ring indicates the dimension of the Ecology indicator; orange, the dimensions of the harvest sector indicator; and beige, the dimensions of the post-harvest sector indicator. On the enabling condition panels, shades of green indicate dimensions comprising different components of the enabling conditions

generally comparable to one another, and to the FPI database average. Important exceptions are the rights and collective action components, where the industrial fishing countries have the capacity to actively participate in the RFMO process and work to institutionalize their access rights, and in some cases harvest rights as well[33,34]. Local market fleets have lower enabling conditions relative to the international fleets and the FPI average in Macro and Governance factors, as well as for use of Access Rights and Data. International fleets' Infrastructure exceeds the FPI average, with scores capturing well organized logistics from offload to final market, while domestic fleets' much lower Infrastructure scores are attributed to various limitations in delivering product that meets international quality requirements, or access to the markets willing to pay the most.

The aggregate performance of each RFMO is largely driven by its constituent markets (Fig. 1c). Harvest sector differences are more pronounced than in Fig. 1a, because local market fleets make up a small part of the total catch of each RFMO, except for the IOTC fisheries where semi-industrial and artisanal fleets represent about half the catch (Supplementary Table 1). Focusing exclusively on meticulously-handled, high value bluefin tuna, CCSBT achieves high post-harvest scores, with lower scores in the Risk dimension driven by wider fluctuations in price and seasonality, and lower Crew performance associated with difficult conditions (12–24 month trips) and low pay for residents of distant water fishing countries with relatively high standards of living (e.g., Japan). The IATTC fisheries scored have strong harvest sector performance, and the post-harvest sector performs well except for the Markets reached. This underscores the influence of trade policy on fishery performance, as the IATTC score reflects US duty waivers for Ecuadorian pouched skipjack during a study-period US ban on Mexican canned tuna. The ICCAT fisheries perform comparably to IATTC in the harvest and post-harvest sectors, though with a lower Stock score reflecting the Atlantic and Mediterranean bluefin stock status during the scoring years. However, with a relatively greater dependence on the high value sashimi market, and the prevalence of European ranching and Asian-owned fishing companies, Captains and Crew dimensions score high, as they are well-compensated and respected in their home communities. Performance of the WCPFC's fisheries mirrors that of the canned fleet in Fig. 1a, as WCPFC's catch is dominated by canned skipjack. Lastly, while the IOTC has reasonably healthy stocks during the scoring years, these fisheries score lower in the harvest sector dimensions because IOTC catch volume is balanced between canned skipjack, which scores above the FPI average, and lower-performing yellowfin and bigeye catch by small-scale fleets throughout the region. While potentially high value, this is not realized because insecure cold chains and inefficient transportation networks funnel these landings to local markets.

Comparing enabling conditions across RFMOs (Fig. 1d), Stock Habitat scores are high, because high seas fisheries are isolated from pollution and disease, even in regions with lower Environment scores. CCSBT, comprised of high-income fishing nations, has the strongest set of scores in the Macro dimension, followed by ICCAT, because the FPI framework scores based on the flag states involved. Most IOTC and IATTC catch is by fleets from countries with weaker governments and economies. Access Rights scores are higher in fisheries with allocated fishing days. The strong individual quotas utilized by CCSBT and for bigeye by ICCAT represent strong individual Harvest Rights. There are comparable levels of Participation across all regions due to the nature of the RFMO management process, but Community scores reflect greater commonality among IATTC fisheries than in other regions. The IATTC has the most Management resources, while the IOTC has the least. The availability and use of Data also range

widely, with the industrialized nations fishing within CCSBT and ICCAT being able to apply more resources to data and analysis than the developing nations that dominate the WCPFC and IOTC member rosters. Infrastructure varies widely as well, with highest levels in CCSBT, slightly ahead of WCPFC, IATTC and ICCAT, and trailed considerably by the IOTC region, where infrastructure limits the ability to preserve harvest value and transport product to high value markets.

When fisheries are grouped by scale of production (Fig. 1e), harvest and harvest assets performance are similar, in contrast to the notable differences among non-tuna fisheries in the FPI database (Supplementary Table 2). However, differences emerge in other harvest sector dimensions. Industrial fleets—those with freezer capacity, mechanized gear, and sophisticated levels of technology onboard—have lower levels of Risk and higher levels of Owners and Captains performance than both semi-industrial fleets—those with mechanized gear, iced holds, and the presence of technology—and the artisanal fleets—those without mechanized gear or built-in cold storage and very limited technology. Semi-industrial and artisanal tuna fisheries have lower Risk scores than non-tuna fisheries, and Owner and Captain performance are comparable at corresponding industrial scales. In the post-harvest sector, artisanal tuna fisheries' post-harvest assets outperform more industrialized tuna fisheries because they capture a premium from maintaining quality fish for high-paying markets, even though the processing involved is not as sophisticated. Processors of industrially-caught longline and purse seine fish have a larger gap between plant managers performance and plant workers performance, because larger operations require managerial capability, but the associated labor is lower-skilled.

The Macro enabling conditions for the industrial fleets (Fig. 1f) reflect the national economies of the flag states, which far exceed those of semi-industrial and artisanal tuna vessels from their respective coastal states. Industrial tuna fleets have coordinated through the RFMO process to establish access rights and secure harvest rights, but not to the same extent as the non-tuna industrialized fleets of the United States, European Union, and New Zealand that make up the comparison set (Supplementary Table 2). Smaller-scale tuna operations use rights-based access far less than their industrialized counterparts, and also less than their non-tuna artisanal counterparts. RFMO engagement is reflected in high scores on dimensions of co-management. The RFMO process offers semi-industrial tuna fisheries a pathway to engage in collective action like industrial fisheries and facilitates Participation by tuna fisheries at all scales of production. This elevates semi-industrial and artisanal tuna fisheries above their comparable non-tuna fisheries. RFMO meetings also support community enabling conditions because they provide a venue for leadership and social cohesion among individuals, even as participants represent their national and corporate interests[7,35]. Tuna vessels have similar Management Inputs as non-tuna vessels across all scales of production, though less use of data and analysis. Artisanal tuna fleets score lower in markets & trade and in Infrastructure conditions than non-tuna artisanal fleets, though semi-industrial tuna fleets are comparable to their non-tuna counterparts. With onboard refrigeration and freezer technology, industrial fleets hold their product until they transship to, or directly land at, ports with infrastructure that supports reliable cold chains, efficient transportation, and international regulatory compliance.

## Discussion

Tuna fisheries represent a significant portion of global fishery volume and value, providing food, key nutrients, and a source of essential income to coastal states whose primary resource is fish.

We characterize where the benefits of 21 major tuna fisheries—representing up to 96% of global tuna production—are accruing, and where there are differences in management and enabling conditions that may affect those outcomes. Our analysis reveals that the biggest variations in performance among tuna fisheries are not found among the RFMOs that manage them, but rather among the final markets that they are able to reach. Harvest sector benefits are comparable across fisheries, as historical open access and leaky limited entry has limited harvester returns to the market return to the capital and labor used. This is reflected in low scores for fisheries with low capital investment and unskilled harvesters, and in moderate scores in capital-intensive fisheries with higher operational skill. On the other hand, there is significant variation in benefits in the post-harvest sector, determined largely by how successfully processors are able to maintain the quality of the catch and distribute it into the highest paying global markets.

This has important implications for current efforts by aid agencies, foundations, and NGOs seeking to identify ways developing coastal states can leverage the tuna stocks in their exclusive economic zones into sustainable sources of food and economic growth. Many of these efforts focus on strengthening coastal states in negotiating better access agreements with harvesting nations, or participating directly in the harvest sector. While fisheries have the potential to provide supernormal returns to the capital invested reflecting resource rent, tuna harvesting demonstrates only moderate performance relative to other fisheries, with differences in benefits among tuna fisheries tracking only the level of capital that must be invested at each scale. This suggests current management is not allowing for the generation of significant resource rent, and therefore that policies designed to capture resource rents from the harvest sector will have limited potential.

Instead, there is more potential value in products currently being sold at low prices in local markets because simple infrastructure and handling practices do not capture or maintain export quality. While such a strategy would export micronutrient-rich protein from developing coastal states[36], this analysis highlights that tuna sold in local markets generates significantly less value for harvesters and processors than that sold in the international canned and sashimi markets. Many poor coastal states face geographic, infrastructure, and capital market disadvantages, and thus are excluded from the benefits of processing. This income, if captured locally through wages, returns to local capital, or resource rents, can be used to support a broader economic base[37], and the opportunity to purchase a more diverse diet on the world market.

However, this analysis illustrates the importance of who is likely to own and work in facilities that improve value, and thus capture benefits from them. A small developing state may not have many highly skilled jobs, such as industrial purse seine captain, but it also lacks the skill base; if needed in a domestic fleet, those jobs would be filled by someone with international training. Similarly, moderate-income states often do not enjoy direct economic or employment benefits from processing plants, as local community members lack the skillset or capital to own or manage the plant, and a migrant workforce is often utilized for unskilled labor positions. While these experiences do not mean that expanding harvest or processing sector development in a particular region will not provide a local boost, they should introduce skepticism about where there are potential gains from the tuna industry.

In addition to identifying broad strategies, these benchmark FPI case studies can be used to identify areas for investment by providing points of empirical comparison, highlighting areas where tuna performs well, as well as opportunities for improvement in fisheries with similar enabling conditions. Within each of the RFMO-level fisheries, there is considerable heterogeneity, and individual countries or fleets can compare themselves to their RMFO fleet average, or to identify peers from which they may be able to transfer best practices.

Once a desired improvement is identified, the FPIs provide a framework for developing a theory of change. Stakeholders can choose outcome measures on which they would like to improve, and the specific enabling factors the project will modify. The FPIs can support the narrative arguing the changed enabling factors lead to the desired improvement in outcomes by providing empirical evidence, drawn from other case studies that have tried similar interventions. A well-supported narrative helps entrain stakeholders and potential funders who wish to distinguish successful approaches.

As the project is implemented, progress through the narrative can be tracked by rescoring the FPIs. While any project should invest most of its monitoring and evaluation budget in data processes that support more precise measurement of its target enabling factors and outcomes, the FPIs' rapid assessment strategy provides an affordable approach to broad-based monitoring beyond those specific project objectives, either for (possibly unintended) changes arising directly from project activity, or for exogenous changes which may affect the project priorities or the narrative supporting the theory of change. Further, the FPIs can be used for ex-post project evaluation of previous investments to identify whether and when particular strategies establish long-term sustainable benefits.

The volume and value of global tuna fisheries represent an important source of both food and income for millions of people, many in low-income or developing countries. While the RFMO system has been largely successful at curtailing overfishing of tuna stocks, healthy stocks have not led to uniformly high economic and social benefits. As with many allocation issues, disagreement about the distribution of those benefits has hindered improvements, and the consensus-driven international resource management bodies are not structured to resolve these issues. The FPIs highlight where fishery benefits are accruing, and where there are opportunities to improve or reallocate them. By drawing on comparison fisheries who have preceded in targeted reforms, it is possible to develop an understanding of what initiatives successfully improve outcomes under similar enabling conditions. This understanding will be used to inform governance, management and market improvement strategies for tuna fisheries, in order to sustain and enhance the environmental, economic, and community welfare benefits generated by them.

## Methods

**Applying the FPIs to tuna fisheries.** The FPIs[27] consist of 68 metrics of fishery outcome performance, grouped into interpretive dimensions capturing stock performance; performance of the harvest sector including harvest levels, the value of harvest assets, risks, and conditions for vessel owners and crew; and performance of the post-harvest sector including market characteristics, processing, the value of processing assets, and conditions for processing owners and workers (Supplementary Figure 1). 54 additional metrics capture enabling conditions which may affect outcome performance, including macroeconomic and governance factors, the strength of property rights, co-management, management resources and methods, and post-harvest infrastructure and markets (Supplementary Figure 2). For each metric, scorers select a point on a 1 to 5 scale, with quantitatively or qualitatively defined boundaries selected to capture variation across global fisheries. While a 5 is interpreted as better performance for outcome measures, it simply captures a higher level (whose effect on performance is uncertain) for enabling condition metrics. Scorers draw on the best available information, including data, proxy data, and local expert knowledge for each metric, and additionally give a confidence grade to each score. Absent systematically quantitative data on many fisheries or fishery sectors of interest, the FPIs represent a relatively robust process for capturing fishery information.

As a standardized tool for assessing fishery performance, the FPIs can facilitate comparisons among tuna fisheries, and between tuna and non-tuna fisheries. However, interpreting these comparisons requires understanding how the FPIs are

applied to the enormous geographical and jurisdictional ranges of the stocks, people, and businesses involved in tuna fishing. While artisanal fleets fish in domestic territorial waters, the majority of global volume is landed by distant water vessels owned in one country (often dictating the nationality of the captain/officer level crew), potentially flagged in another, fishing in multiple countries' territorial waters or the high seas, with crew hired from an altogether different region of the world. To assess performance consistently, the FPI lens adopts the perspective of the businesses and people who participate directly in the fisheries. Each fleet we scored was associated with its flag state (e.g. for governance scores or to assess enforcement capacity), and therefore flag states within an RFMO's jurisdiction were considered as local stakeholders of the relevant management body. Within the harvest sector, we attempt to capture outcomes for tuna fishing only, excluding vessels that do not fish, such as support or shipment. Therefore, in applying the FPI framework to tuna, the post-harvest sector includes first receiver processing upon shore delivery (e.g. heading and gutting or finning for bullets are still considered harvest practices for large tunas).

For industrial fleets, the notion of fishing community participation in management in the FPI framework was interpreted to encompass the people who represented stakeholder interests at regular RFMO meetings. For measures that reference fishery participants' local opportunities, the target person is evaluated relative to their home; e.g. captains and crews of distant water fleets are more meaningfully compared to their homes, even though their residence may be geographically distant from the point of harvest or processing.

While the FPIs measure whether and how direct participants in the industry capture fishery benefits, they omit benefits coastal states receive from access payments paid by harvesting countries. These payments are made to coastal state governments that often do not directly participate in harvesting or processing tuna, and thus their outcomes are not captured within the FPI measures of harvest or post-harvest sector outcomes. For example, access agreements can include provisions for schools or roads. We use access payments as a partial basis for scoring the value of harvest asset rights measure, however we do not assess the non-fishery benefits that arise from them. Linking benefits from payments required within fishing access agreements, and associated diplomatic and aid arrangements, with any particular support for fishing or non-fishing people or communities would be extremely difficult. Although an important effect of tuna fishing, and sometimes a significant component of gross domestic product for small coastal states (e.g. Kiribati), these payments are typically small relative to the value of the harvest, leading to controversy about fair division of rents between distant water fleets and coastal states. By systematically examining where benefits accumulate within the fishery, our analysis highlights how participating businesses and communities are performing in order to identify effective strategies for coastal states to receive appropriate benefits from their tuna resources.

**Major desktop scoring resources**. The 21 major tuna fisheries were scored through assembling existing literature and reports, interviews with key informants in science, management, and industry, and visits to key tuna landing locations. Many fisheries are heterogeneous and/or widely distributed. In these cases, we selected representative fleets within a fishery which were individually scored (e.g. the Indian Ocean artisanal longline/handline fishery is comprised of the scored domestic fleets of four countries). A fishery score, however, is the volume-weighted average of individual fleet scores, with adjustments to reflect known non-representativeness of the directly scored fleets.

For all fisheries, stock assessment information was taken from the 2012 ISSF summary report. Economic and community information was gathered through in-person assessment for all IOTC fleets, IATTC purse seine and longline fleets, and WCFPC small-scale fleets. There was sufficient literature to desk-score other fleets, supported by expert interviews. Hamilton et al. [29] was particularly valuable in characterizing global industrial value chains, and Gillett[28] provided essential information on small-scale fisheries. Metric scores and notes are included in the supplementary materials.

Based on 2009 volumes, we directly scored fleets comprising 63% of global landings (italicized in Table 1). That number increases to roughly 77% of the world's tuna production when we include the volume of fleets that were not scored directly, but where we are confident our scored fleets are representative (non-italicized in Table 1); e.g. the unscored WCPFC Korean purse seine fleet is similar to the WCPFC Taiwanese and Japanese purse seine fleets we did score. An additional 19% of production is attributed to fleets that fall into fisheries we scored, but where we lack the information to be confident in associating their performance with a scored fleet. However, additional information could extend our assessment to as much as 96% of global production. The remaining 4% of volume is comprised of either very small fisheries, or fisheries lacking desk-scoring resources.

**Data availability**
Metric scores for the 21 aggregate tuna fisheries are available in Supplementary Data 1.

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

## Acknowledgements

This work was funded by PROFISH group, The World Bank. We thank Kirsten Anderson of Turnstone Design for our figures. We are grateful to the many people who contributed time and expertise in helping us understand the fisheries reported.

## Author contributions

J.L.A. and C.M.A. developed the methodology; J.K.M. and C.M.A. collected the data and conducted the analysis; C.M.A. and J.K.M. wrote the article.

## Additional information

**Competing interests:** The authors declare no competing interests.

