## [Peer Review File · Nature Communications]

Reviewers' comments:

Reviewer #1 (Remarks to the Author):

I think this paper is very well written and is a novel application of a methodology that gathers existing information and informed viewpoints about fishery performance in a systematic way. I was familiar with the Fishery Performance Indicators from an earlier paper that lays out the methods. However, I did not fully appreciate the potential of this approach until seeing it applied to a category of fisheries in a more detailed way.

What is done especially well in this paper is the discussion around the interpretation of the indicator scores. This is done in two ways: 1) clearly demonstrated expertise about global tuna fisheries is used to shed light on why scores came out as they did, and 2) insightful interpretation of what the scores may be indicating that may not have been known before.

There is one area that I think needs more clarity: in the last full paragraph of page 7, you raise the issue of comparing non-tuna FPIs with tuna FPIs given the “enormous geographical and jurisdictional ranges of the fish stocks, people, and businesses involved” in tuna fisheries. It seems as if you are saying that these comparisons should not be made yet later in the paper there are such comparisons. Are you saying that the issues you highlight (artisanal fleets vs distant water, ships flagged in one country with crew from another, etc) make comparisons among tuna fisheries challenging? You mention strategies used to correct for this. Your discussion of this point is a bit unclear.

Further, if there truly are reasons to think that scores could be systematically different in tuna vs non-tuna fisheries then perhaps those comparisons should be put into a separate section of the paper. In so doing, the comparisons could still be made but caveated that the difference in scores could be due, in part, to factors beyond what a particular indicator is trying to measure.

I am attaching the Word version of your document with a few very minor suggested edits using track changes.

Reviewer #2 (Remarks to the Author):

This paper presents an assessment of Fishery Performance Indicator scores for global tuna fisheries. The authors provide a carefully crafted picture of the state of global tuna fisheries across not only ecological indicators, but also importantly social and economic dimensions. These results serve as a useful benchmark that can be used to target and assess the impacts of management interventions. As is somewhat inevitable for a paper of this scope, it lacks a clear set of striking results (the clearest being that the largest distinction in FPI scores comes at the scale of final market, rather than management body), but the sheer scale and careful crafting of the exercise makes this an important contribution to the field of fisheries science. Specifically, while we are gaining an increasing understanding of the ecological state of global fisheries, very little data exists on the socio-economic performance of fisheries at a large scale, and those studies that do exist are often not easily comparable to each other. I therefore believe the presented work is both original (in that no one to my knowledge has provided this broad reaching an assessment of the state of tuna fisheries) and significant (by providing a benchmark example of the use of the FPIs that others can emulate).

From the perspective of data and methods I have not substantial comments. As a large, complex, and mostly qualitative process reproducibility is somewhat of a concern, but the authors present the underlying data in the appendix so that readers with specific questions about results can dig into the raw-er numbers.

My only substantial critique of the paper is that it reads at times like a recitation of various numbers and comparisons, which by its nature it is, without a clear narrative to the results. While this is somewhat inevitable outcome of a broad survey like this, I would recommend at providing a clearer set of figures to help readers digest the massive amount of information presented here. I find Figure 1 to be too cluttered to be visually effective, see specific comments below. Again from the perspective of communication, while Figure 1 contains just about every result in the paper in one spot, I would recommend the authors pick 1-2 of the most striking results and create figures dedicated to illustrating those points. Are there geographic trends in the data that could be mapped? What regions have the most to be gained by for example switching from local market to a canned or sashimi grade market?

On the whole this is a carefully researched, well written, and important study and I recommend publication with minor revisions.

A few specific notes below:

Table.1: Define scale term in caption, as well as significance of colors

Line 134: Typo "dayss"

Line 192: "In order to"

Line 199: define "heading and gutting"

Line 218: typo: "our analysis will highlight industry sectors how..."?

Line 247: "to be viewed"

Fig.1's resolution is too low, unless a higher-res version is going to be used for the print version. I personally find these figures to be too busy. The main goal as I take it is to be able to see each of the scores, and compare them to the mean in the background, and at the moment that's not what pops out to me; I have to constantly go back and forth from the figure to the legends to the caption to interpret what I'm seeing. The background plotting for the different categories is too subtle to be of much use but noticeable enough to be very visually distracting. I would recommend instead color coding the text of the indicators (e.g. "stock health" could be green for ecological, "harvest", "harvest assets" etc some other color.

Line 289: In a similar vein to the comments on figure one, these are some very esoteric color choices; I'm not sure that "taupe" and "putty" will be instantly recognizable colors for most readers

Reviewer #3 (Remarks to the Author):

The authors conclude that "the biggest variations in performance among tuna fisheries are not found among the RFMOs that manage them, but rather among the final markets that they target." Given that some of the information used is subjective or only semi-quantitative (e.g. based on interviews), the authors should explain how robust this conclusion and the entire analyses are.

The manuscript is a valuable contribution but it would benefit from improved writing (there are many awkward sentences and uncommon acronyms are used extensively) and more references to support the assertions made.

Also, I had a difficult time throughout the text figuring out when the present tense refers to the 2012 baseline, and when it refers to today. A careful revision to address this would be useful.

Reviewer #1 (Remarks to the Author):

I think this paper is very well written and is a novel application of a methodology that gathers existing information and informed viewpoints about fishery performance in a systematic way. I was familiar with the Fishery Performance Indicators from an earlier paper that lays out the methods. However, I did not fully appreciate the potential of this approach until seeing it applied to a category of fisheries in a more detailed way.

What is done especially well in this paper is the discussion around the interpretation of the indicator scores. This is done in two ways: 1) clearly demonstrated expertise about global tuna fisheries is used to shed light on why scores came out as they did, and 2) insightful interpretation of what the scores may be indicating that may not have been known before.

Thank you for your supportive words! This is exactly what we are hoping to achieve with this paper. The FPI team is working on several papers which demonstrate several different ways the tool can be used.

There is one area that I think needs more clarity: in the last full paragraph of page 7, you raise the issue of comparing non-tuna FPIs with tuna FPIs given the “enormous geographical and jurisdictional ranges of the fish stocks, people, and businesses involved” in tuna fisheries. It seems as if you are saying that these comparisons should not be made yet later in the paper there are such comparisons. Are you saying that the issues you highlight (artisanal fleets vs distant water, ships flagged in one country with crew from another, etc) make comparisons among tuna fisheries challenging? You mention strategies used to correct for this. Your discussion of this point is a bit unclear. Further, if there truly are reasons to think that scores could be systematically different in tuna vs non-tuna fisheries then perhaps those comparisons should be put into a separate section of the paper. In so doing, the comparisons could still be made but caveated that the difference in scores could be due, in part, to factors beyond what a particular indicator is trying to measure.

Thank you for pointing out this apparent inconsistency. We do not intend to say that these comparisons cannot or should not be made (as you point out, we make them). Rather, interpreting the details of the comparisons requires understanding how the FPIs are applied to tuna fisheries, which is what the rest of this section goes on to clarify. The most important distinction is that in most of our other applications of the FPIs, we discuss how a fishery contributes to a community, with the implication that it is a community of place. Most tuna fisheries, on the other hand, have participants from, and distribute benefits to, many different communities of place. Therefore, we do not think of a fishery contributing to the health of a single community in the same way. In order to clarify this, we have reworded the introduction to this paragraph, and reminded the reader of this distinction when we introduce results on community.

I am attaching the Word version of your document with a few very minor suggested edits using track changes.

Thank you for these specific suggestions. We have revised the text to reflect them.

Reviewer #2 (Remarks to the Author):

This paper presents an assessment of Fishery Performance Indicator scores for global tuna fisheries. The

authors provide a carefully crafted picture of the state of global tuna fisheries across not only ecological indicators, but also importantly social and economic dimensions. These results serve as a useful benchmark that can be used to target and assess the impacts of management interventions. As is somewhat inevitable for a paper of this scope, it lacks a clear set of striking results (the clearest being that the largest distinction in FPI scores comes at the scale of final market, rather than management body), but the sheer scale and careful crafting of the exercise makes this an important contribution to the field of fisheries science. Specifically, while we are gaining an increasing understanding of the ecological state of global fisheries, very little data exists on the socio-economic performance of fisheries at a large scale, and those studies that do exist are often not easily comparable to each other. I therefore believe the presented work is both original (in that no one to my knowledge has provided this broad reaching an assessment of the state of tuna fisheries) and significant (by providing a benchmark example of the use of the FPIs that others can emulate).

Thank you for highlighting this contribution. This is what we are hoping to accomplish with this paper. And, indeed, this was a massive effort to assemble information from published (and grey literature) studies, as well show up on the ground to get information in places where such studies do not exist.

From the perspective of data and methods I have not substantial comments. As a large, complex, and mostly qualitative process reproducibility is somewhat of a concern, but the authors present the underlying data in the appendix so that readers with specific questions about results can dig into the rawer numbers.

We bristle a little at the characterization of the FPIs as a qualitative process: most measures have an underlying quantitative score system. Absent quantitative data to which to apply the score system, we ask experts with information on each fishery to place it within predetermined ranges of the quantitative structure. We are sanguine about the fact that this is less more dependent on judgement than systematically collected, rigorously quality controlled primary data. Since that doesn't exist in many cases (we do use it where available), we still believe conscientious efforts using the best information available lead to some valuable insights. In the long run, our hope is that methods such as ours will facilitate comparisons and progress in improving the livelihoods associated with fisheries, while at the same time inducing fisheries to create processes that enable us to replace information with data.

To make this clearer to readers, and in response to a similar comment from another reviewer, we have added some additional statements about how we try to maximize the robustness of the method in describing the FPI process.

My only substantial critique of the paper is that it reads at times like a recitation of various numbers and comparisons, which by its nature it is, without a clear narrative to the results. While this is somewhat inevitable outcome of a broad survey like this, I would recommend at providing a clearer set of figures to help readers digest the massive amount of information presented here. I find Figure 1 to be too cluttered to be visually effective, see specific comments below. Again from the perspective of communication, while Figure 1 contains just about every result in the paper in one spot, I would recommend the authors pick 1-2 of the most striking results and create figures dedicated to illustrating those points. Are there geographic trends in the data that could be mapped? What regions have the most to be gained by for example switching from local market to a canned or sashimi grade market?

This tension between providing a comprehensive overview and a more detailed analysis is definitely something with which we wrestled in developing the manuscript. In presenting this work, we found that different categories of users, each with their own perspective, want to draw different things out of this

analysis. Thus, we opted for a relatively raw presentation of the data so different users could see themselves in the figure, and make the comparison of greatest interest to them. These include RFMO representatives who, naturally, want to compare performance by RFMO; coastal state representatives with different raw products who want guidance on capturing more fishery benefits; industry representatives in different sectors who want to compare themselves on other sectors; small and large scale operators who want to compare themselves to one-another. Picking any particular result would necessarily compromise some users' abilities to do this.

Representing this information on a map is an interesting suggestion for future work, as there are regional differences which different data could disaggregate into subregional or national level (by coastal state, flag country, etc.). Certainly the greatest volumes that could be moved from the local market to the sashimi market through better handling are in the Indian Ocean region. At the level of this global analysis, however, we hesitate to impose a geographic focus on this conclusion because we think the general point that there is value to be captured by improving handling to move product out of low value markets applies anywhere.

While we continue to feel our current figures best achieve the broad audience goals of this paper—and the journal's length cap doesn't let us add more figures and discussion—your comment also highlighted difficulty identifying our narrative. While there are myriad results, we are hoping readers come away struck by the fact that differences in economic outcomes (across market, and to a lesser extent, industrial scale) arise in the post-harvest sector, while harvest sector outcomes are similar, reflecting normal returns to capital. RFMO effects are just expressing differences in the products and markets accessed by each RFMO. This is important because it indicates that the post-harvest sector for high value products is where coastal states can act to increase their local benefits from fisheries. Our presentation is trying to be respectful of the journal's natural science norm that interpretation is saved for discussion, but we have added some additional guidance for readers directing them toward these patterns in the data. In the Discussion, we offer interpretation, integration and implications. For our analysis, an advantage of this is it allows us to synthesize across the different ways we grouped the fleet.

On the whole this is a carefully researched, well written, and important study and I recommend publication with minor revisions.

A few specific notes below:

Table.1: Define scale term in caption, as well as significance of colors

Done.

Line 134: Typo "dayss"

Thank you.

Line 192: "In order to"

Thank you.

Line 199: define "heading and gutting"

Spelled out abbreviation, thanks.

Line 218: typo: “our analysis will highlight industry sectors how...”?

Thank you.

Line 247: “to be viewed”

Thank you.

Fig.1’s resolution is too low, unless a higher-res version is going to be used for the print version. I personally find these figures to be too busy. The main goal as I take it is to be able to see each of the scores, and compare them to the mean in the background, and at the moment that’s not what pops out to me; I have to constantly go back and forth from the figure to the legends to the caption to interpret what I’m seeing. The background plotting for the different categories is too subtle to be of much use but noticeable enough to be very visually distracting. I would recommend instead color coding the text of the indicators (e.g. “stock health” could be green for ecological, “harvest”, “harvest assets” etc some other color.

We worked with a professional designer to develop this information-dense figure. We’ve consulted with her to ensure the resolution of the final version will be better; it helps that the journal takes final figures as separate files (the initial submission double-processed a pdf, which did not do good things to the crispness of the lines or text). In this revision, we also worked with her to reduce the background color distraction by instead coloring only the outer ring of each radar graph. We had not tried this format, but we think it is an improvement.

Line 289: In a similar vein to the comments on figure one, these are some very esoteric color choices; I’m not sure that “taupe” and “putty” will be instantly recognizable colors for most readers

We’d actually made a wager over this word choice. We have changed this to reflect your suggestion, and one author now owes another a beer.

Reviewer #3 (Remarks to the Author):

The authors conclude that "the biggest variations in performance among tuna fisheries are not found among the RFMOs that manage them, but rather among the final markets that they target." Given that some of the information used is subjective or only semi-quantitative (e.g. based on interviews), the authors should explain how robust this conclusion and the entire analyses are.

We have added some additional discussion of robustness in the introduction of the FPI tool.

The manuscript is a valuable contribution but it would benefit from improved writing (there are many awkward sentences and uncommon acronyms are used extensively) and more references to support the assertions made.

We have reread the manuscript and made a number of editorial changes to smooth the text, in addition to taking some specific suggestions from other reviewers. We found most of our claims were already supported by citations or were conclusions from our own analysis; in some cases our assertions are synthesizing across references for individual fisheries, in which case their supporting citations are represented in the Supplementary Material.

Also, I had a difficult time throughout the text figuring out when the present tense refers to the 2012 baseline, and when it refers to today. A careful revision to address this would be useful.

This is a good point, and one to which we were not adequately attentive in the previous version. We had been mostly consistent in using past tense to refer to researcher decisions on study design and present tense to refer to results, per journal policy to have results of current policy in present tense (based on our c.2012 data). We were not adequately careful in identifying features we knew to no longer be true today (2018). We have gone through the manuscript in an effort to call out when present tense makes statements that are not true today.

REVIEWERS' COMMENTS:

Reviewer #2 (Remarks to the Author):

Thank you for your careful response to my comments (and glad that I could help resolve an outstanding bet!). The nature of scoring methods are clearer (and I acknowledge that my use of the term "qualitative" was loose in my prior review). The narrative of the paper is much clearer while maintaining the broad range of information desired by the authors. The figures in particular are much easier to read and I feel now present a clearer picture of the results. This is a truly impressive synthesis of a massive global industry.

Reviewer #3 (Only provided remarks to the Editor)